# Operational Efficiency of Mexican Water Utilities: Results of a Double-Bootstrap Data Envelopment Analysis

**Jose Humberto Ablanedo-Rosas [1], Aaron Guerrero Campanur [2], Elias Olivares-Benitez [3,*], Jacqueline Y. Sánchez-García [4] and Juan Enrique Nuñez-Ríos [4]**

[1] Marketing and Management Department, University of Texas at El Paso, 500 W. University Avenue, El Paso, TX 79968, USA; jablanedorosas2@utep.edu
[2] TecNM/ITS de Uruapan, Tecnologico Nacional de Mexico Carr. Uruapan-Carapan 5555, Uruapan, Michoacan 60015, Mexico; aaronguerrero@tecuruapan.edu.mx
[3] Facultad de Ingeniería, Universidad Panamericana, Alvaro del Portillo 49, Zapopan Jalisco 45010, Mexico
[4] Escuela de Ciencias Economicas y Empresariales, Universidad Panamericana, Alvaro del Portillo 49, Zapopan Jalisco 45010, Mexico; jsanchezg@up.edu.mx (J.Y.S.-G.); jnunezr@up.edu.mx (J.E.N.-R.)
* Correspondence: eolivaresb@up.edu.mx; Tel.: +52-33-1368-2200

**Abstract:** The objective of this paper is to estimate the operational efficiency of Mexican water utilities and identify the context variables that impact their efficiency. In particular, a bootstrap data envelopment analysis (DEA) and a bootstrap truncated regression analysis are combined in a two-stage research method. In the first stage, an input-oriented DEA model is used to determine bootstrap efficiency scores. Then, the corrected distribution function of the efficiency scores is used to estimate a truncated regression which is aimed to identify the significant influential context variables. Three categorical and two continuous context variables are considered in the analysis. Results show that only one context variable has a significant impact on the water utilities efficiency scores. Managerial recommendations are drawn from the analysis. It is suggested that water utilities continue or implement wastewater treatment, persist in decreasing and controlling leakage across the distribution network, and maximizing sewer coverage.

**Keywords:** water utilities; water management; data envelopment analysis; bootstrap data envelopment analysis; double-bootstrap approach

---

## 1. Introduction

The significant and undesirable effect of economic and social activities on all natural resources is a very active current research topic. Human activity is modifying the hydrologic cycle creating unexpected climate changes that originate severe droughts and floods. A vital resource for the subsistence of the human race is drinking water. Drinking water management is considered a significant metric when investigating national economic and sustainable development and social welfare [1–3]. Developing countries show low operational efficiency of water and sewerage utilities, which leads to serious environmental and health issues [4]. These issues exacerbate with the rapid development of urban areas. This article investigates the management of this vital resource in the developing country of Mexico.

The Organization for Economic Co-operation and Development (OECD) estimates a world population of more than 10 billion people for the year 2050, which translates to a water demand increase of about 55% [3]. If the potable water consumption continues at the same rate, it is predicted that two-thirds of the world population will face water supply issues in the year 2025 [5]. Participants

in the World Economic Forum in Latin America, held in Brazil in 2018, identified the efficient supply and management of water as the fifth most important challenge faced by Latin American countries.

Mexico covers 1.96 million km$^2$; it is the fifth and thirteenth largest country in the Americas and the world respectively. It has an estimated population of 121 million people. The Mexican National Water Commission (Comisión Nacional del Agua, CONAGUA) forecasts that climate change will unevenly affect different geographic areas thus generating a huge contrast between dry and wetland regions, which constitutes a major threat for water availability [6].

The rainy season in central Mexico lasts roughly from May through October. The rain is heaviest from June to September in the southeast region of the country. The yearly rainfall is about 1489 billion m$^3$; around 73%, 22%, and 6% of the rainfall evaporates, flows to the rivers, and infiltrates the subsoil to recharge the aquifer mantles respectively [6]. This unused rainwater suggests that water availability in central Mexico has not yet reached a critical level. However, water availability is characterized by multiple asymmetries. The distribution of water is unbalanced and intermittent resulting in poor distribution, restricted access, and uneven rates. In this regard, Kauffer [7] states that water availability does not entirely explain the problem of drinking water access in Mexico. Hence, it is even more important to analyze the adequate use and management of this vital resource.

A frequent stream of research in water management is the privatization of water utilities as a strategic solution for water supply improvement. It is expected that private water utilities could be more efficient than public entities. These scenarios have been widely studied, drawing no conclusion. The relationship between the type of ownership and efficiency is unclear in the water distribution sector [4,8–14].

Data envelopment analysis is a very popular technique when investigating the efficiency of water utilities. Some examples showing the variety of studies and approaches are as follows. De Witte and Marques [15] used the length of mains and number of employees as inputs, and the volume of delivered drinking water and the number of connections as outputs when assessing the water utilities efficiency among five different countries. They concluded that a technological gap among countries could explain some inefficiencies. In the case of a geographic region, Sáez-Fernández et al. [16] investigated the technical performance of water utilities under the sustainability perspective. They analyzed the case of the water industry in the region of Andalusia, Spain. The water utilities supply potable water and provide sewerage services. They highlighted the importance of a few critical performance metrics to assess the sustainability of the water industry. The key metrics are as follows: the volume of water introduced in the pipelines, the volume of water consumed by customers, and the volume of water lost by leakage. In a different research objective, González-Gómez et al. [11] tested the impact of the type of ownership on the performance efficiency of water utilities. They analyzed private, public, and private-public water utilities in Spain. Their study of 80 rural water companies did not provide any conclusion about the relationship between types of ownership and operational efficiency. More recently, Vishwakarme et al. [17] investigated water supply in an Indian state. They identified factors of inefficiencies and analyzed their mitigation through public policies and regulation. In a broad perspective, results from DEA have been used to recommend infrastructure, regulatory, and managerial improvements in water and sewerage systems [7,18–24].

This paper is aimed at analyzing the operational efficiency of water utilities in Mexico by means of a double-bootstrap DEA approach. The remaining part of this paper is organized as follows. Section 2 presents a literature review. The proposed research methodology is introduced in Section 3. Section 4 presents the analysis of the water utilities in Mexico and discusses results. Conclusions are given in Section 5.

## 2. Literature Review

The literature review is aimed at identifying the current research gaps in the current state of the art. There are many studies about water utilities management and a high percentage of them studied their efficiencies. However, just a few studies utilized an approach combining DEA and bootstrapping,

and only one addressed the Mexican water utilities. Therefore, this literature review is split into three sections: the first one is a peculiar analysis of water efficiency studies by means of a bibliometric and main path analysis, the second section analyses the studies combining bootstrapping, DEA, and regression methods, and the last one reviews the case of the Mexican water utilities.

### 2.1. Bibliometric and Main Path Analysis

Bibliometrics was introduced by Garfield in 1972 [25,26]. Modern bibliometrics uses automated methods and applications to explore, organize, and analyze large volumes of historical data that can be used in strategic decision-making processes [27].

This research uses Clarivate Analytics' Web of Science database as its data source. The following phrases were utilized to identify the research articles: "Water" AND "Data Envelopment Analysis" OR "Stochastic Frontier Analysis". The word "Water" was defined in the title field. DEA and SFA were selected as topic that includes title, keywords, and abstract. Notice that this search framework is not exhaustive. It provides just information of papers using either DEA or SFA for assessing water issues. The aim of this section is to present a quick descriptive overview of research activities and trends in the aforementioned search framework. The reader is referred to Berg and Marques [28] for a thorough discussion of a literature survey in the field.

The Web of Science database was accessed on 3 February 2019 and the search identified a total of 192 research articles and 6 proceeding papers from 1993 to 2019. Figure 1 shows the steady increase of research articles in the field. The first research paper was published in 1993. Between 1993 and 2005 there were several years without any publications. However, starting in 2005, the rate of growth of research articles shows a non-linear increasing trend.

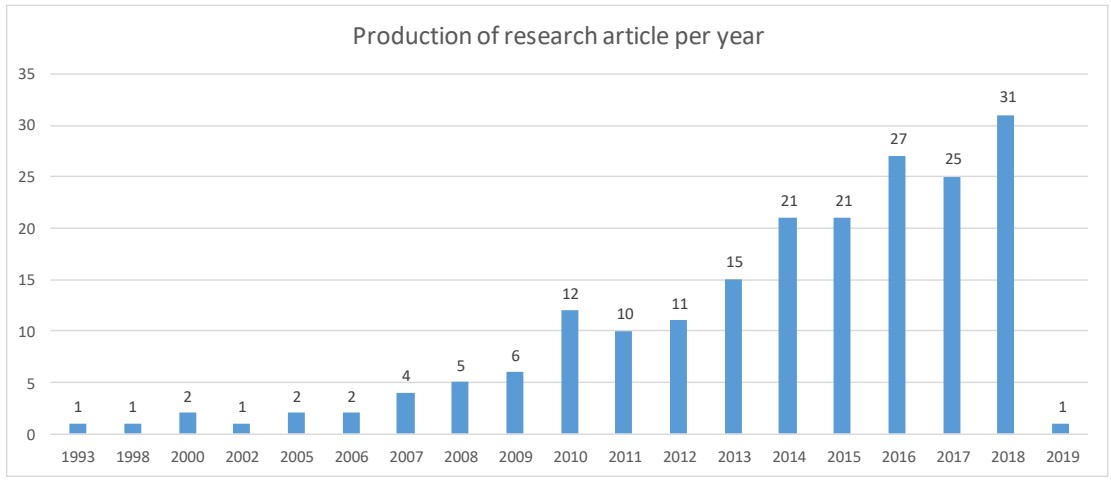

**Figure 1.** Number of research article per year.

The top eleven leading authors, by number of documents published, with at least 5 publications are listed in Table 1. The top author is Maria Molinos-Senante who is a professor at the Pontificia Universidad Catolica de Chile. The second author is Ramon Sala-Garrido who is a professor at the Universidad de Valencia, and who collaborates with Molinos-Senante. The collaborative work between Molinos-Senante and Sala-Garrido highlights the importance of social research networks.

Citations, in which one article refers to earlier research publications, are the standard means by which researchers acknowledge the source of their methods, ideas and findings. Additionally, paper's citations are often used as a rough estimate of a paper's significance. Table 2 presents the top five cited papers. Kirkpatrick et al. [12] is the most cited paper with just under one hundred citations. Kirkpatrick et al. [12] studied the privatization effect of water services in African countries. Their results did not show evidence that private utilities perform better than public utilities. Their arguments

for this lack of improvement are the technology of water supply, the nature of the product, transaction costs, and regulatory weaknesses.

**Table 1.** Most productive authors (with at least 5 publications).

| Author | Number of Articles |
|---|---|
| Molinos-Senante M | 21 |
| Sala-Garrido R | 18 |
| Guerrini A | 11 |
| Marques RC | 11 |
| Romano G | 11 |
| Speelman S | 8 |
| Frija A | 6 |
| Gonzalez-Gomez F | 6 |
| Buysse J | 5 |
| Maziotis A | 5 |
| Van Huylenbroeck G | 5 |

**Table 2.** Most cited papers.

| Paper | Total Citations |
|---|---|
| Kirkpatrick C. et al. [12] The World Bank Economic Review | 97 |
| De Witte K, and Marques R. C. [29] Central European Journal of Operations Research | 87 |
| Thanassoulis E. [30] European Journal of Operations Research | 78 |
| Aida K., et al. [31] Omega The International Journal of Management Science | 74 |
| Thanassoulis E. [32] European Journal of Operations Research | 66 |

Another interesting metric to consider is the classification of countries by numbers of publications during the study period. Table 3 shows the publication productivity of countries in the research stream. The total number of articles published ranks the countries. The most productive country is China with 37 publications. There is a significant gap with the next country, which is the USA. Table 3 shows 11 countries, of which two are from Asia, one from North America, two from South America, five from Europe, and one from Oceania. Table 3 suggests a huge heterogeneity of cases and studies when researching water utilities performance.

**Table 3.** Most productive countries.

| Country | Number of Articles |
|---|---|
| China | 37 |
| USA | 20 |
| Chile | 17 |
| Italy | 16 |
| Spain | 14 |
| Belgium | 10 |
| India | 10 |
| Australia | 9 |
| Portugal | 8 |
| Brazil | 6 |
| Germany | 6 |

Several journals have a great deal of influence in the research stream (Table 4). The top two research outlets are focused on water issues. The third one necessarily includes water issues as well. The fourth outlet has a broader aim and covers all issues associated to sustainability; it is good to know that water utilities performance is being studied from the sustainable perspective.

**Table 4.** Most relevant sources.

| Source | Number of Articles |
|---|---|
| Water Policy | 14 |
| Water Resources Management | 14 |
| Utilities Policy | 11 |
| Sustainability | 8 |
| Journal of Cleaner Production | 7 |
| Water | 7 |
| Applied Economics | 6 |
| Environmental Science and Pollution Research | 6 |
| Agricultural Water Management | 5 |
| Journal of Productivity Analysis | 5 |
| Journal of Water Resources Planning and Management | 5 |
| Water Science and Technology: Water Supply | 5 |

The analysis of keywords helps to understand the content of the 198 published articles. Table 5 shows a high focus on efficiency and reveals a higher preference among researchers for DEA over SFA. It suggests that most studies conducted in the field have been performed in the Chinese water industry.

**Table 5.** Most relevant keywords.

| Author Keywords | Frequency |
|---|---|
| Data Envelopment Analysis | 95 |
| Efficiency | 32 |
| Water utilities | 21 |
| Technical efficiency | 17 |
| Stochastic frontier analysis | 14 |
| Water use efficiency | 14 |
| Benchmarking | 13 |
| Performance | 13 |
| China | 10 |
| Water supply | 9 |

A research article having more than one author is called a co-authored article and the authors are said to be co-authors of each other. An author collaboration network (Figure 2) is one having authors as nodes and an edge between two authors if both have co-authored a research article. The co-authorship network analysis is used to identify strong collaborative ties between researchers. Figure 2 indicates five clusters with strong collaboration. The strongest cluster is the green one showing the collaboration between Molinos-Senate, Sala-Garrido, and Maziotis. On other hand, the largest cluster is the orange one depicting the collaboration between Speelman, Buysse, Chebil, Frija, and Van Huylenbroeck.

Researchers in scientific areas cite each other in their research articles, mainly in the journals devoted to their respective areas. Such citations form a directed network through which there are many influential paths. The main paths through such citation networks contain the key intellectual developments in these scientific fields. Figure 3 depicts these ideas using the citations network defined when searching the research topics "water", "data envelopment analysis", and "stochastic frontier analysis" in the web of science database. Figure 3 depicts the main path and the relevant publications that correspond to the influential papers about the aforementioned topics. The most cited papers do not necessarily define the Main Path [33].

Figure 3 shows 21 influential papers that were identified among 198 articles extracted from the Web of Science database, through the main path analysis using the key-routes technique. They constitute the main subjects discussed in the research topics. Figure 3 depicts a clustered network, each node constitutes one research paper, the size of the node is proportional to the number of citations of the article, and the edge indicates the relationships among articles. The arrowhead shows the direction of the flow

within the main path. A unique color identifies each cluster and indicates a major subject in the research stream. The cluster's subject was determined by a deep analysis of the articles defining the cluster.

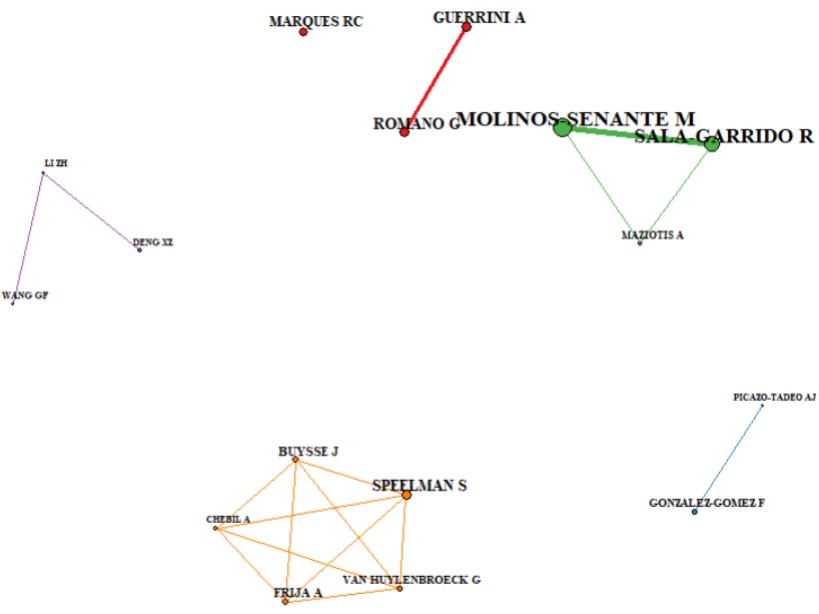

**Figure 2.** Author collaboration network.

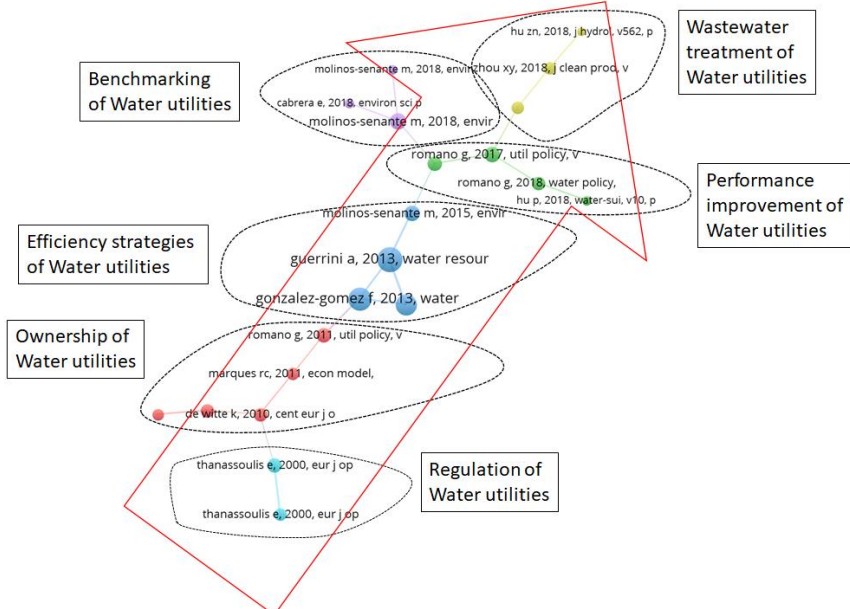

**Figure 3.** Main path suing the global key route.

The main path analysis (Figure 3) shows a total of six clusters. The main path begins at the light blue cluster with Thanassoulis [32] pointing to Thanassoulis [30]. The light blue cluster derives in the red cluster, which points to the blue cluster, which supports the green cluster. The green cluster defines two lines of thoughts that derived in the purple and yellow clusters where the main path ends with Molinos-Senante and Farías [34] and Hu et al. [35] respectively. There are 198 articles in the Web of Science discussing water issues using either DEA or SFA as research methods. The main path identifies the 21 papers that constitute the backbone of the knowledge published in this specific research stream for a period of about 25 years.

### 2.2. Bootstrap DEA

This paper uses double-bootstrap DEA as research method. Sometimes this method is called a two-stage DEA approach. In general, the first stage corresponds to a bootstrap data envelopment analysis and the second stage is associated with a bootstrap truncated regression analysis. The purpose of stage one is to estimate the distribution function of the efficiency scores, while the reason of the second stage is to identify the environmental or context variables that significantly influence the efficiency scores. Hence, this approach combines a non-parametric analysis with a parametric method. This section analyzes the papers that investigate water utilities with a bootstrap DEA approach.

Anwandter and Ozuna [4] used DEA and regression analysis to the case of the Mexican water utilities; they concluded that privatization and other public sector reforms did not positively impact efficiency. Recently, Wang et al [23] estimated the water use efficiency in 30 Chinese provinces. They used sewage as undesirable output and incorporated the analysis of five environmental variables. They found that export dependence, technological progress, and educational value have positive influence in water services efficiency. On the other hand, industrial influence and government intervention have negative impacts and no influence in water services efficiency, respectively. These studies did not apply to any bootstrap procedure. Table 6 shows the inputs, outputs, and environmental variables considered in these two papers.

**Table 6.** Papers using DEA and regression analysis.

| Paper | Inputs | Outputs | Environmental Variables |
|---|---|---|---|
| Anwandter and Ozuna [4] | Personnel Electricity Materials Chemical Outside services Other costs | Water supply Primary treatment Secondary Treatment | State or municipal water utility Autonomous regulation Service cut allowed Water lost/water produced Population density Non-residential users Technological progress |
| Wang et al. [23] | Labor Capital Water | Sewage Per capita GDP | Government intervention Education Industrial structure Export |

Simar and Wilson [36] argue that this two-stage DEA approach is incorrect and leads to an inappropriate inferential process. They suggest the use of a bootstrap procedure to overcome this drawback. The initial efficiency scores are not a good estimator of the true efficiency scores' distribution. Hence, a bootstrap procedure of the efficiency scores is useful to infer the real distribution of efficiency. Moreover, the bootstrap DEA provides the unbiased or corrected efficiency scores. Pawsey et al. [37] used a bootstrap DEA; they used four different models, and all of them had operating cost as a common input and the number of water connections as a unique output. Historic cost, regulatory asset base, and physical asset base were the distinct inputs used in models one, two, and three respectively. They concluded the new public management reforms had a significant impact on 16 Victorian water utilities.

This paper uses the bootstrap approach recommended by Simar and Wilson [36] A total of 16 papers using a similar approach to assess the performance of water utilities were identified. They span from 2007 to 2018 and are listed with their inputs, outputs, and context variables in Table 7.

The papers listed in Table 7 are all similar in the sense that they use a variant of a bootstrap DEA approach to study water utilities, but they are different in the scope, objective, and the results of the study. The difference in the analytical approach begins with the selection of inputs, outputs, and context variables, and it ends with the particular mathematical procedure to combine DEA, bootstrapping, and regression analysis.

The scope of studies varies from countries to states. De Witte and Marques [29,38], Halkos and Tzeremes [1], and Mbuvi et al. [39] address the assessment of water services among countries. Ananda [40],

Carvalho and Marques [41], Guerrini et al. [42], lo Storto [43], Marques et al. [44], Molinos-Senante et al. [45], Pinto et al. [46], Pointon and Mathews [21], and Zschille and Walter [47] investigate the performance of water utilities in a specific country. Finally, Güngör-Demirci et al. [48], Güngör-Demirci et al. [49], and Renzetti and Dupont [50] focus their studies on the water utilities of one state.

These research articles identified the context variables that have a significant impact on efficiency scores [40,43,44,47,50]. Carvalho and Marques [41], Pinto et al. [46], and De Witte and Marques [29,38] found that regulatory and benchmarking incentives significantly impact efficiency. A positive impact from clean drinking water and adequate sewage disposal was identified by Halkos and Tzeremes [1] and Zschille and Walter [47]. However, Carvalho and Marques [41] concluded that the water utilities' performance is lower when drinking water supply and wastewater services are offered. Güngör-Demirci et al. [48] determined that precipitation had a negative impact on efficiencies; but in a second different study, Güngör-Demirci et al. [49] concluded a positive and negative impact of precipitation in a financial and production model respectively. Ananda [40], Güngör-Demirci et al. [49], and Molinos-Senante et al. [45] highlighted the positive impact of customer density; however, the results of Guerrini et al. [42] showed the opposite. Mbuvi et al. [39] concluded that the country's economic development is positively linked to utilities efficiency. Geographical location was a significant positive factor in the study of Lo Storto [43]. These results show a broad heterogeneity that is similar to the diversity of the analyzed cases. De Witte and Marques [15] discuss a special methodology based on metafrontier analysis to assess the impact of environmental variables.

The extensive diversity of analyzed cases and their results confirm the complexity of defining a standard set of inputs, outputs, and environmental variables for assessing the efficiency of water utilities. This study adds to this complexity by characterizing the current situation of water utilities in Mexico.

**Table 7.** Papers using bootstrap DEA and regression analysis.

| Paper | Inputs | Outputs | Environmental Variables |
|---|---|---|---|
| De Witte & Marques [29,38] | Number of employees<br>Length of mains | Volume of water<br>Number of connections | Leakage<br>Groundwater extraction<br>Industry water/household delivery<br>Gross regional product<br>Water unique activity<br>Corporatization<br>Delivery in one municipality<br>Regulator<br>Benchmarking<br>Elevation |
| Renzetti & Dupont [50] | Materials<br>Labor<br>Distribution length | Sum of annual deliveries | Population density<br>Residential water usage/total water agency output<br>Surface or groundwater<br>Private dwellings<br>Summer temperature<br>Precipitation<br>Scope (combinations of water & wastewater)<br>Ownership<br>Regulation |
| Carvalho & Marques [41] | Staff cost<br>Operations and maintenance expenses<br>Capital expenses | Volume of water delivered<br>Number of customers | % Purchased water<br>% of surface water provided<br>% surface water source<br>Customer density-water<br>Customer density-wastewater<br>Peak factor<br>% Residential customers |

**Table 7.** *Cont.*

| Paper | Inputs | Outputs | Environmental Variables |
|-------|--------|---------|------------------------|
| Halkos & Tzeremes [1] | Gross fixed capital formation (% of GDP) Labor force | GDP | Pop. with sustainable access to water Pop. with sustainable access to sanitation |
| Mbuvi et al. [39] | Employees Network Length | Population served Water sold Total water connections Daily water supply Pop. served/Target pop. Water sold/Target pop Water connections/Target pop. | Independent regulation Performance contract use GDP Network density |
| Zschille & Walter [47] | Revenues | Water meters Water delivered to households Water delivered to nonhouseholds Network length Population Volume of water intake | Output density Leak ratio Groundwater ratio Elevation difference Debt per capita Dummy for east Dummy for private Dummy for sewage |
| Guerrini et al. [42] | Depreciation + interest paid Staff costs Operating costs Length of mains | Population served Total revenues | Degree of investment diversification Customer density Size |
| Lo Storto [43] | Aqueduct network length Sewerage network length Total production cost | Revenue from service delivered | Number of municipalities Number of connections Population Num. of connections/total network length Num. of connections/num. of municipalities % surface water % recycling water % groundwater |
| Ananda [40] | Operating expenditure Length of water mains | Total urban water supplied Output quality | Total connected properties Properties served per km of water main % residential consumption Leak Production density Region Prefecture Owner Water source Vertical integration Peak factor |
| Marques et al. [44] | Capital cost Staff cost Other operational expenditures | Volume of water billed Number of customers | Consumption per capita Customer density Water losses Monthly water charge Outsourcing Subsidies Gross domestic product (GDP) Time |

**Table 7.** *Cont.*

| Paper | Inputs | Outputs | Environmental Variables |
|---|---|---|---|
| Pointon & Mathews [21] | Labor Capital Other | Water delivered Equivalent population served | Water abstraction from rivers Total water pop./length of mains Total sewerage pop./length of sewers Leakage Trade effluent |
| Pinto et al. [46] | Mains length Staff Other operational costs | Volume of water sold Number of households | Different types of water sources Vertical integration of the services Economies of scope Corporatization Private sector participation, Customer density Economies of scale Household disposable income Desired quality of service. |
| Güngör-Demirci et al. [48] | Operating expenses Energy | Operating revenue | Number of connections Customer density Groundwater volume/total water production Leak Precipitation |
| Güngör-Demirci et al. [49] | Operating expenses Energy | Financial model: Operating revenue Production Model: Volume of water sold | Number of connections Customer density Groundwater volume/total water production Nonrevenue water Precipitation |
| Molinos-Senante et al. [45] | Operating costs Labor Network length | Water distributed Customers with wastewater treatment service Indicator of drinking water quality | Non-revenue water Peak factor Customer density Ownership Water source |

## 2.3. Mexican Water Utilities

The water operation agencies (Organismos de operacion del agua) are the entities responsible for water supply and sewerage services in Mexico. These agencies are responsible for efficiently supplying good-quality potable water and maintaining the sustainability of this vital resource. However, the actions of these agencies are focused on increasing the water system infrastructure. There is scarce information about water quality, service quality, and performance indicators [51].

Anwandter and Ozuna [4] studied the case of the Mexican water utilities. Their study was motivated by the worldwide efforts to decentralize public utilities as an improvement strategy in developing countries. The water utility sector was not exempted from these public reforms efforts and they analyzed the reforms' impact in the context of the Mexican water utilities.

Since its development, a national company operated the water system in Mexico. The first decentralization attempts initiated in the 1950s with the creation of potable water committees at the municipal level. It is in the 1980s when control was passed from the federal government to the state and municipal level.

Anwandter and Ozuna [4] used Mexican utilities data from 1995. Their DEA model considered the following inputs: number of employees, number of kilowatt-hours consumed, materials cost, chemical cost, outside service cost, and other costs. Water supply, primary treatment, and secondary treatment were the corresponding outputs.

Anwandter and Ozuna [4] found about 50% of water suppliers efficient. In a second step, Anwandter and Ozuna [4] used two methods to compare the impact of environmental variables (public sector reforms) in the efficiency scores of the Mexican water utilities. Six environmental variables composed by three categorical and three continuous variables were used in the second stage. The three categorical variables correspond to: 1) a variable indicating if the water utility was a state or municipal entity, 2) a variable signaling if the water utility was or was not an autonomous regulator, and 3) a variable showing if the ceasing of service was allowed (to minimize the undesirable effect of late payments). The three continuous variables are: 4) the ratio between water lost and water produced, 5) population density, and 6) percent of non-residential users.

The results of Tobit regression showed that two variables were significant, the ratio of water lost to water produced which had a negative effect on the efficiency, and the percentage of non-residential users, which had a positive effect on efficiency. They concluded that public sector reforms did not have a significant impact on the water utilities efficiencies.

The decentralization of water utilities to the municipal level or toward autonomous water operators did not have a positive impact on the operational efficiency of Mexican water utilities. The water sector has a monopolistic nature; the introduction of competition could help to alleviate the operational asymmetries in the sector and support their performance improvement [4].

This paper contributes in all the subsections of this literature review section. Considering the bibliometric and main path analysis subsection, this research article supports the increasing nonlinear trend of water efficiency studies worldwide. It contributes to the large variety of real cases analyzed in the research literature. It expands the study of the impact of wastewater treatment in the efficiency of water utilities, as suggested by the arrowhead in the main path (Figure 3). Section 2.2 reviews the studies using bootstrap DEA as research method. This paper contributes with a different selection and assessment of inputs, outputs, and context variables. Furthermore, the outputs selected are current performance measures being tracked and reported by Mexican water utilities. Finally, associated with the last subsection, this paper updates the study of the efficiency of Mexican water utilities. The previous study analyzed the impact of public reforms implemented about 20 years ago. After several years of reform implementation, this paper is aimed at assessing the operational efficiency of these water utilities and at identifying the significant context variables that influence the aforementioned operational efficiency.

## 3. Proposed Methodology

DEA was first introduced by Charnes et al. [52] and has been developed into a widely accepted academic field. Thirty-some years after the publication of the seminal paper, the use of DEA to analyze relative efficiency continues and does not show any signs of weakening [53].

A decision-making unit (DMUs) represents the entity under assessment; in this research a DMU represents a utility that supplies drinking water. Each DMU has an associated set of inputs and outputs respectively, which represent multiple resources and performance measures. Consider a set of n DMUs, each $DMU_j$ (j = 1, ... , n) consumes m inputs $x_{ij}$ (i = 1, 2, ... , m) for producing s outputs $y_{rj}$ (r = 1, 2, ... , s). The relative efficiency of a particular $DMU_0$ is defined as a ratio of the weighted sum of outputs to the weighted sum of inputs, and is obtained by solving the following fractional programming problem:

$$max \frac{\sum_{r=1}^{s} u_r y_{r0}}{\sum_{i=1}^{m} v_i x_{io}}$$

Subject to

$$\frac{\sum_{r=1}^{s} u_r y_{rj}}{\sum_{i=1}^{m} v_i x_{ij}} \leq 1, \quad j = 1, 2, \ldots \ldots, n \tag{1}$$

$$u_r \geq 0, \quad r = 1, 2, \ldots \ldots, s$$

$$v_i \geq 0, \quad i = 1, 2, \ldots \ldots, m$$

where $u_r$ is the weight given to th*e r*th output and $v_i$ is the weight given to the *i*th input. The fractional program can be converted into a linear programming problem where $w_0$ is the relative efficiency of $DMU_0$, the DMU under evaluation.

$$w_0 = max \sum_{r=1}^{s} u_r y_{r0}$$

Subject to

$$\sum_{i=1}^{m} v_i x_{io} = 1,$$

$$\sum_{r=1}^{s} u_r y_{rj} - \sum_{i=1}^{m} v_i x_{ij} \leq 0, \quad j = 1, 2, \ldots \ldots, n \tag{2}$$

$$u_r \geq 0, \quad r = 1, 2, \ldots \ldots, s$$

$$v_i \geq 0, \quad i = 1, 2, \ldots \ldots, m$$

In this model, the weighted sum of the inputs for $DMU_0$ is forced to be 1, and $DMU_0$ is efficient if and only if $w_0 = 1$; a score less than 1 implies that $DMU_0$ is inefficient.

The corresponding dual formulation of model (2) is given by

$$min \theta_0$$

Subject to

$$\sum_{j=1}^{n} \lambda_j x_{ij} \leq \theta_0 x_{i0}, \quad i = 1, 2, \ldots . m$$

$$\sum_{j=1}^{n} \lambda_j x_{rj} \leq y_{r0}, \quad r = 1, 2, \ldots . s \tag{3}$$

$$\lambda_j \geq 0, \quad j = 1, 2, \ldots . . n$$

where $\lambda_j$ (*j* = 1, . . . , *n*) are nonnegative scalars and $\theta_0$ is the efficiency of $DMU_0$, the DMU under assessment. Model (3) is known as the CCR model [52] and is an input oriented model with constant returns to scale assumption.

$$\sum_{j=1}^{n} \lambda_j = 1 \tag{4}$$

When Equation (4) is added to model (3), the BCC model [54] is defined and it corresponds to an input oriented model with variable returns to scale assumption.

The solution of model (3) assigns the value of 1 to all efficient DMUs, making it difficult to differentiate among all efficient DMUs. Several approaches have been proposed for differentiating DMUs when there are more than one efficient DMU. A common approach is the so called super efficiency DEA model introduced by Andersen and Petersen [55]. This method enables efficient DMUs to achieve an efficiency score greater than one, facilitating the assignment of rankings to all efficient DMUs.

An extension of DEA is the cross efficiency method which was developed for identifying the best performing DMUs and for ranking DMUs using cross efficiency scores [56,57]. The advantage of the cross efficiency method is that it alleviates the weak discrimination of the classical DEA model. The cross efficiency method has two steps. In the first step, the classical efficiency scores are determined using model (3) with either constant or variable returns to scale assumption. A set of optimal weights preserving the efficiency values for each $DMU_p$ is determined in the second step, and these weights are used for calculating the peer evaluation score $\theta_{pj}$ of $DMU_j$ (j = 1, . . . , n) using the weights obtained

for $DMU_p$. Once all peer evaluation cross-efficiency scores have been calculated, each $DMU_j$ has n cross-efficiency scores. The overall cross-efficiency score X-Eff$_j$ for each specific $DMU_j$ is determined by estimating its corresponding mean of cross-efficiency scores.

The distribution function of the efficiency scores is unknown. Simar and Wilson [36] introduced a method to bootstrap the DEA scores and estimate the unknown distribution of the efficiency values. In a second stage, the method evaluates the impact of context variables in the efficiency scores. The Simar and Wilson [36] estimation method is described as follows:

Step 1. Compute the estimated efficiency score for each $DMU_j$ (j = 1, ... , n) by solving model (3). The linear program has to be solved n times and at each solution make $\theta_j = \theta_0$ to obtain the efficiency score for each $DMU_j$.

Step 2. Compute a truncated maximum likelihood estimation to regress the efficiency scores against the context variables, $\theta_j = z_j\beta + \varepsilon_j$ (j = 1 to n), and provide an estimate $\hat{\beta}$ of the coefficient vector $\beta$ and estimate $\hat{\sigma}_\varepsilon$ of $\sigma_\varepsilon$, the standard deviation of the residual errors $\varepsilon_j$.

Step 3. For each $DMU_j$ (j = 1, ... , n) replicate the following steps (3.1 to 3.4) a total of $B_1$ times. This process will generate $B_1$ bootstrap estimates $\hat{\sigma}_{jb}$ of $\sigma_j$ (j = 1 to n, b = 1, ... , $B_1$).

Step 3.1 Generate the residual error $\varepsilon_j$ from the normal distribution $N(0, \hat{\sigma}_\varepsilon^2)$.

Step 3.2 Estimate the regression $\theta_j^* = z_j\hat{\beta} + \varepsilon_j$ (j = 1 to n).

Step 3.3 Produce a pseudo data set of inputs and outputs as follows: $x_j^* = x_j\left(\frac{\theta_j}{\theta_j^*}\right)$ and $y_j^* = y_j$ (j = 1 to n).

Step 3.4 Use the pseudo data set $(x_j^*, y_j^*)$ to estimate the pseudo efficiency score $\hat{\theta}_j^*$ by solving model (3).

Step 4. Estimate the bias corrected efficiency $\hat{\theta}_j$ for each $DMU_j$ (j = 1, ... , n) using the bootstrap estimator or the bias $\hat{b}_j$ where

$$\hat{\theta}_j = \theta_j - \hat{b}_j \text{ and } \hat{b}_j = \left(\frac{1}{B_1}\sum_{b=1}^{B_1}\hat{\theta}_{jb}^*\right) - \theta_j.$$

Step 5. Compute a truncated maximum likelihood estimation to regress the bias corrected efficiency scores against the context variables, $\hat{\theta}_j = z_j\beta + \varepsilon_j$ (j = 1 to n), and provide an estimate $\hat{\beta}^*$ of the coefficient vector $\beta$ and estimate $\hat{\sigma}_\varepsilon^*$ of $\sigma_\varepsilon$, the standard deviation of the residual errors $\varepsilon_j$.

Step 6. Replicate the following steps (6.1 to 6.3) a total of B2 times. This process will generate B2 pairs of bootstrap estimates $\left(\hat{\beta}_j^{**}, \hat{\sigma}_{\varepsilon j}^{**}\right)$ (b = 1, ... , B2).

Step 6.1 Generate the residual error $\varepsilon_j$ from the normal distribution $N(0, \hat{\sigma}_\varepsilon^{*2})$.

Step 6.2 Estimate the regression $\hat{\theta}_j^{**} = z_j\hat{\beta}^* + \varepsilon_j$ (j = 1 to n).

Step 6.3 Compute a truncated maximum likelihood estimation to regress $\hat{\theta}_j^{**}$ against the context variables and provide an estimate $\hat{\beta}^{**}$ of the coefficient vector $\beta$ and an estimate $\hat{\sigma}_\varepsilon^{**}$ of $\sigma_\varepsilon$, the standard deviation of the residual errors $\varepsilon_j$.

Step 7. Construct the $(1 - \alpha)$% confidence interval for vector $\beta$.

Therefore, in this study a double-bootstrap DEA approach is utilized to estimate the efficiency of the Mexican water utilities and to assess the impact of five context variables in their performance. A total of $B_1 = 2000$ and $B_2 = 4000$ bootstrap replications were calculated at the first (steps 1–4) and second stage (steps 5–7) of the double-bootstrap DEA approach, respectively.

## 4. Case Study

Several studies in developing countries have pointed out that scarcity and availability of data is a major issue [19,21,22]. Furthermore, inaccurate data make the process even harder [18]. Mexico is not the exception of this trend. This study is based on the available data of water utilities in Mexico. The data were obtained from the Measuring Program for Management of Operating Organizations

(Programa de Indicadores de Gestion de Organismos Operadores) which is part of the Mexican Institute of Water Technology (Instituto Mexicano de Tecnologia del Agua) and correspond to year 2016. This research analyzes 36 (n = 36) major organizations responsible for supplying potable water across the country of Mexico. These water suppliers operate in 21 out of 32 federal states and 12 out of 13 hydrological-administrative regions. Three inputs, three outputs, and five context variables were gathered for each DMU. This configuration satisfies the rule of thumb that $n \geq \max(m \times n, (m + n) \times 3)$.

The sets of inputs, outputs, and context variables are part of the public performance report of the Mexican water utilities. Some of these performance indicators are ratios and their consideration in our research is unavoidable. Firstly, the scarcity of data impedes the transformation of ratios into absolute measures. Secondly, the unbalanced and uneven characteristics of drinking water distribution in Mexico suggest the use of ratio data. For example, if water distribution is intermittent and variable, then the volume of distributed water is a biased measure of consumption of this resource. While on the contrary, the ratio liters/per capita/day provides a better estimation of the volume of water delivered to consumers. Issues associated with ratio data and DEA are discussed by Hollingsworth and Smith [58], Emrouznejad and Amin [59], and Hatami-Marbinia and Toloob [60]. For examples of research articles addressing water utilities efficiency using ratio data, the reader is referred to [12,39,61,62].

## 4.1. Inputs

The first input is the volume of water fed in the water distribution system. It is listed as water distribution and it is distinct from water consumption. The difference between volume of water consumption and water distribution is explained by leakage and unused water. Water distribution is measured in liters per consumer per day and it is aligned with the studies of Renzetti and Dupont [50], Carvalho and Marques [41], Mbuvi et al. [39], Zschille and Walter [47], Ananda [40], Pointon and Mathews [21], and Molinos-Senante et al. [45]. The number of employees per thousand consumers is the second input. It is, perhaps, the most commonly used input when studying the efficiency of water utilities. The number of employees was considered by De Witte and Marques [15,29], Renzetti and Dupont [50], Carvalho and Marques [41], Mbuvi et al. [39], Guerrini et al. [42], and Pointon and Mathews [21]. Zschille and Walter [47] used total revenue as a proxy for total costs. They argued that revenue covers all operational costs. This is true for the Mexican water utilities, which depend on revenue and monetary subsidies from the government for covering operational expenses. We use accounts with on-time payment as proxy for revenue. Furthermore, this approach could be used for reviewing "whether tariffs for water deliveries are reasonable, and if not, by how much they can be reduced." Zschille and Walter [47], p. 3757. Hence, accounts with on-time payment is the third input and is a proxy of the ratio water-sold/target-population reported by Mbuvi et al. [39]. Since account with on-time payment is the source of income, it is a proxy for revenues [42,43,47–49] and capital [21,23,24,38].

## 4.2. Outputs

The outputs correspond to real performance measures reported by the water utilities in Mexico. They are financial-operational ratios and they have to be adequately considered when used in DEA formulations [63]. The first output is the ratio of water production cost to water volume produced. This performance measure should be minimized. Efficient DMUs should have a lower ratio than inefficient DMUs. Hence, this ratio is normalized accordingly. This performance measure falls into line with the inputs and outputs reported by De Witte and Marques [29,38], Carvalho and Marques [41], Mbuvi et al. [39], Guerrini et al. [42], Ananda [40], Güngör-Demirci et al. [48,49], and Molinos-Senante et al. [45]. The ratio of volume of water paid to volume of water produced is the second output. Efficient DMUs should have a higher ratio than inefficient DMUs. This performance measure is aligned with the inputs and outputs proposed by De Witte and Marques [29,38], Carvalho and Marques [41], Mbuvi et al. [39], Zschille and Walter [47], and Güngör-Demirci et al. [49]. The financial ratio of total expenses to total income is the third output. The direction of this output is the same as the first output. Efficient



DMUs should have a lower ratio than inefficient DMUs. Hence, it is normalized accordingly as well. Carvalho and Marques [41], Zschille and Walter [47], Guerrini et al. [42], Güngör-Demirci et al. [48], and Güngör-Demirci et al. [49] used inputs and outputs in concordance with this performance measure.

### 4.3. Context Variables

The context variables are exogenous factors that affect the efficiency of DMUs. Tables 6 and 7 show a great variety of variables tested in different cases of water utilities. The results are not conclusive. A variable that could inhibit efficiency in a specific case could enable efficiency in a different environment. The reason is the broad heterogeneity of water utilities' characteristics; they have different internal and external features that define specific behaviors. Hence, the case of Mexican water utilities is a particular one. In the past years, several water utilities have migrated from a system of a fixed fee for residential customers to a system of metering consumer's consumption. It is expected that metering, at both the macro and micro level, will enable better control of the system and will support better performance. Still, not all Mexican water utilities perform either macro-metering or micro-metering. Hence, water macro-metering and water micro-metering were considered as context variables to study their impact on the efficiency of Mexican water utilities. Water macro-metering and water micro-metering are two different categorical variables. Each categorical variable has a value of 1 if it has been implemented by the water utility, otherwise it is 0. Zschille and Walter [47] considered water meters in their study.

Several authors identified a significant relationship between efficiency of water utilities and wastewater treatment [29,38,40,41,45]. This synergy suggests the presence of economies of scope [38,42] when a water utility simultaneously distributes drinking water and treats wastewater. Therefore, wastewater treatment is incorporated in this study as a categorical variable; its value is 1 if the water utility performs wastewater treatment, its value is 0 otherwise.

Leakage has an inherent presence in water distribution systems. It is due to multiple factors and consumes a large amount of water companies' resources. De Witte and Marques [29,38], Carvalho and Marques [41], Mbuvi et al. [39], Zschille and Walter [47], Ananda [40], and Güngör-Demirci et al. [48,49] have analyzed its impact on efficiency. This study considers volume of water lost per connection as a continuous context variable. Efficient DMUs should have a lower volume of water lost per connection than inefficient DMUs. Hence, this context variable is normalized accordingly; its inverse value corresponds to the number of connections that account for the loss of one cubic meter of water. Therefore, higher values of this normalized continuous context variable should support higher efficiencies.

The last context variable is sewer coverage. Zschille and Walter [47] used sewage services as a categorical variable. In this case, sewer coverage is a continuous variable that suggests that companies providing both potable water and sewage services are more efficient than utilities providing just one service, an effect of economies of scope [38]. This idea is aligned with the studies of De Witte and Marques [29,38], Carvalho and Marques [41], Halkos and Tzeremes [1], Lo Storto [43], Pointon and Mathews [21], and Molinos-Senante et al., [45].

### 4.4. Results and Discussion

Table 8 presents the summary of attributes for year 2016. Wastewater treatment is the only variable that presents a standard deviation value that is close to the numerical quantity of the mean, indicating a high variability of wastewater treatment among the water utilities.

DEA can be performed using either a CRS [52] or a VRS [54] assumption. The VRS efficiency score measures pure technical efficiency, whereas the CRS efficiency score is composed of a combination of scale and technical efficiencies; the former is due to the conditions under which the DMU is operating and the latter is due to the operation of the DMU itself. The ratio of CRS to VRS determines the scale efficiency.

**Table 8.** Summary of Attributes.

| Attribute | Units | Min | Mean | Max | Std. Dev. |
|---|---|---|---|---|---|
| *Inputs* | | | | | |
| Water distribution | liters/per capita/day | 149.42 | 249.99 | 400.18 | 63.92 |
| Number of employees per thousand consumers | number of employees | 2.13 | 5.02 | 14.58 | 2.39 |
| Accounts with on-time payment | % | 3.64 | 62.09 | 94.00 | 21.46 |
| *Outputs* | | | | | |
| Ratio production cost/volume produced | $/M$^3$ | 3.56 | 7.57 | 14.45 | 2.67 |
| Ratio water volume paid/water volume produced | % | 7.63 | 45.33 | 79.00 | 16.02 |
| Ratio Total expenses/Total income | % | 70.75 | 95.64 | 183.62 | 20.41 |
| *Context Factor* | | | | | |
| Water macro-metering | Dummy binary variable | 0.00 | 0.81 | 1.00 | 0.40 |
| Water micro-metering | Dummy binary variable | 0.00 | 0.81 | 1.00 | 0.40 |
| Wastewater treatment | Dummy binary variable | 0.00 | 0.50 | 1.00 | 0.51 |
| Volume of water lost per connection | M3/connection | 40.27 | 126.86 | 302.83 | 64.38 |
| Sewer coverage | % | 59.00 | 91.91 | 100.00 | 9.25 |

Table 9 shows the efficiency scores. The second column corresponds to the initial estimation of the DEA-CRS efficiency scores. The third column shows the bootstrap CRS efficiency scores which are the corrected CRS efficiency scores. The range of the bootstrap CRS efficiency scores is [0.4226,0.8967] with a mean of 0.7249. Thus, the water utilities would need an average reduction of 27.51% in inputs to achieve the efficiency level of their most efficient water company. The fourth and fifth column show the initial and bootstrap corrected VRS efficiency scores respectively. The range of the bootstrap VRS efficiency scores is [0.5443,0.9674] with mean of 0.8132. Hence, water suppliers would need an average reduction of 18.68% in inputs to achieve the efficiency level of their most efficient water utility. This DEA-VRS result is based on pure technical efficiency which is translated into management skills [64]. The improvement opportunities suggested by these approaches could be explained as follows. In the case of the first input, water distribution, the result suggests that a big percentage of water is lost through the water distribution systems. This is an inherent characteristic of water distribution systems in Mexico and needs to be appropriately addressed in order to increase efficiency. A decrease in the loss of water across the distribution network will decrease the need for water production while keeping the same rate of water consumption. The number of employees per thousand consumers (second input) is translated into labor sub-utilization. The expected result is a significant increase in outputs associated with an increase in labor productivity. The third input corresponds to the percentage of accounts with on-time payment; it is similar to the second input. A significant increase in output is expected when improving the payment system, which translates to a higher rate of accounts with on-time payment. Furthermore, the average corrected scale efficiency score of 0.8870 score suggests that an increase in the scale of operations will benefit most of the water utilities.

Table 10 shows the confidence intervals of the bootstrap efficiency scores with $\alpha = 0.05$. The water utilities with the higher CRS and VRS efficiency scores are CAASIM and CESPT respectively. They constitute the most attractive water utilities for benchmarking purposes. A t-test is used to test the difference between the bootstrap VRS and bootstrap scale efficiencies; a p-value < 0.01 suggests that both technical and scale factors should be considered to attain efficiency.

**Table 9.** Efficiency and Bootstrap Efficiency Scores.

| | CRS Eff Score | Bootstrap CRS Eff Score | VRS Eff Score | Bootstrap VRS Eff Score | Scale Eff Score | Bootstrap Scale Eff Score |
|---|---|---|---|---|---|---|
| **DMU** | $\theta_j$ | $\hat{\theta}_{jb}$ | $\theta_j$ | $\hat{\theta}_{jb}$ | | |
| COMAPA-G | 0.7770 | 0.7083 | 0.8536 | 0.8183 | 0.9103 | 0.8656 |
| SAPAS-LP | 1.0000 | 0.8619 | 1.0000 | 0.9065 | 1.0000 | 0.9508 |
| SIMAS-PN | 0.7941 | 0.7179 | 0.9659 | 0.9161 | 0.8221 | 0.7836 |
| CESPM | 1.0000 | 0.8707 | 1.0000 | 0.9219 | 1.0000 | 0.9444 |
| DAPA | 0.6895 | 0.6338 | 0.6959 | 0.6589 | 0.9909 | 0.9618 |
| JAPAC | 0.7138 | 0.6467 | 0.8266 | 0.7965 | 0.8636 | 0.8119 |
| JUMAPA | 0.8394 | 0.7430 | 0.8623 | 0.8093 | 0.9734 | 0.9182 |
| OOMAPAS | 0.5362 | 0.4854 | 0.5829 | 0.5443 | 0.9199 | 0.8918 |
| SIMAPAG | 1.0000 | 0.8548 | 1.0000 | 0.9091 | 1.0000 | 0.9403 |
| SOAPAMA | 0.7034 | 0.6464 | 0.7446 | 0.6964 | 0.9446 | 0.9282 |
| AGUAH | 0.9181 | 0.8089 | 1.0000 | 0.8344 | 0.9181 | 0.9695 |
| AMD | 1.0000 | 0.8238 | 1.0000 | 0.8301 | 1.0000 | 0.9923 |
| CAASIM | 1.0000 | 0.8967 | 1.0000 | 0.9142 | 1.0000 | 0.9809 |
| CAAMTROH | 0.8721 | 0.7779 | 1.0000 | 0.9371 | 0.8721 | 0.8301 |
| CAEV | 0.5743 | 0.5191 | 0.6757 | 0.6458 | 0.8500 | 0.8038 |
| CMAPS | 1.0000 | 0.8104 | 1.0000 | 0.8417 | 1.0000 | 0.9628 |
| CESPT | 1.0000 | 0.8933 | 1.0000 | 0.9674 | 1.0000 | 0.9235 |
| COMAPA-R | 0.7840 | 0.7046 | 0.8321 | 0.7788 | 0.9423 | 0.9048 |
| COMAPA-EM | 0.7556 | 0.6547 | 0.8092 | 0.7512 | 0.9338 | 0.8715 |
| CMAS | 0.4733 | 0.4226 | 0.5766 | 0.5447 | 0.8209 | 0.7759 |
| DAPASCH | 1.0000 | 0.8875 | 1.0000 | 0.9319 | 1.0000 | 0.9524 |
| JAPAM | 1.0000 | 0.8341 | 1.0000 | 0.9148 | 1.0000 | 0.9117 |
| JIAPAZ | 0.6424 | 0.5895 | 0.8202 | 0.7839 | 0.7832 | 0.7521 |
| SADM | 0.8720 | 0.7872 | 0.9902 | 0.9140 | 0.8806 | 0.8613 |
| SAPASNIR | 1.0000 | 0.8139 | 1.0000 | 0.9276 | 1.0000 | 0.8775 |
| SAPACG | 0.8550 | 0.7441 | 0.9500 | 0.8540 | 0.9000 | 0.8713 |
| SAPAS | 0.9181 | 0.8290 | 1.0000 | 0.8922 | 0.9181 | 0.9291 |
| SACMEX | 0.4821 | 0.4256 | 0.6197 | 0.5850 | 0.7779 | 0.7276 |
| SIAPASF | 0.9633 | 0.8716 | 0.9809 | 0.9239 | 0.9820 | 0.9433 |
| SMAPA | 0.5033 | 0.4492 | 0.5796 | 0.5475 | 0.8683 | 0.8204 |
| SIMAPARG | 1.0000 | 0.8818 | 1.0000 | 0.9432 | 1.0000 | 0.9349 |
| SIMAPACO | 1.0000 | 0.8276 | 1.0000 | 0.9211 | 1.0000 | 0.8985 |

**Table 9.** *Cont.*

| DMU | CRS Eff Score $\theta_j$ | Bootstrap CRS Eff Score $\hat{\theta}_{jb}$ | VRS Eff Score $\theta_j$ | Bootstrap VRS Eff Score $\hat{\theta}_{jb}$ | Scale Eff Score | Bootstrap Scale Eff Score |
|---|---|---|---|---|---|---|
| SIMAS-A | 0.6814 | 0.6088 | 0.6851 | 0.6394 | 0.9947 | 0.9521 |
| SOSAPAMIM | 0.5537 | 0.4987 | 0.6737 | 0.6424 | 0.8219 | 0.7763 |
| SOAPAP | 1.0000 | 0.8461 | 1.0000 | 0.9208 | 1.0000 | 0.9188 |
| SOSAPAZ | 0.8066 | 0.7222 | 0.9637 | 0.9103 | 0.8370 | 0.7934 |
| Min | 0.4733 | 0.4226 | 0.5766 | 0.5443 | 0.7779 | 0.7276 |
| Mean | 0.8252 | 0.7249 | 0.8802 | 0.8132 | 0.9313 | 0.8870 |
| Max | 1.0000 | 0.8967 | 1.0000 | 0.9674 | 1.0000 | 0.9923 |
| Std. dev. | 0.1784 | 0.1455 | 0.1512 | 0.1317 | 0.0731 | 0.0717 |

**Table 10.** Bootstrap efficiency scores and their confidence intervals.

| DMU | CRS Eff Score $\theta_j$ | Bootstrap CRS Eff Score $\hat{\theta}_{jb}$ | Lower Bound $\hat{\theta}_{jb}$ | Upper Bound $\hat{\theta}_{jb}$ | VRS Eff Score $\theta_j$ | Bootstrap VRS Eff Score $\hat{\theta}_{jb}$ | Lower Bound $\hat{\theta}_{jb}$ | Upper Bound $\hat{\theta}_{jb}$ |
|---|---|---|---|---|---|---|---|---|
| COMAPA-G | 0.7770 | 0.7083 | 0.6541 | 0.7861 | 0.8536 | 0.8183 | 0.7877 | 0.8630 |
| SAPAS-LP | 1.0000 | 0.8619 | 0.7620 | 0.9933 | 1.0000 | 0.9065 | 0.8316 | 1.0906 |
| SIMAS-PN | 0.7941 | 0.7179 | 0.6580 | 0.7969 | 0.9659 | 0.9161 | 0.8725 | 0.9949 |
| CESPM | 1.0000 | 0.8707 | 0.7743 | 1.0554 | 1.0000 | 0.9219 | 0.8576 | 1.0940 |
| DAPA | 0.6895 | 0.6338 | 0.5952 | 0.6768 | 0.6959 | 0.6589 | 0.6287 | 0.7008 |
| JAPAC | 0.7138 | 0.6467 | 0.5991 | 0.7291 | 0.8266 | 0.7965 | 0.7748 | 0.8395 |
| JUMAPA | 0.8394 | 0.7430 | 0.6772 | 0.8171 | 0.8623 | 0.8093 | 0.7669 | 0.8742 |
| OOMAPAS | 0.5362 | 0.4854 | 0.4490 | 0.5223 | 0.5829 | 0.5443 | 0.5201 | 0.5740 |
| SIMAPAG | 1.0000 | 0.8548 | 0.7513 | 0.9830 | 1.0000 | 0.9091 | 0.8362 | 1.0784 |
| SOAPAMA | 0.7034 | 0.6464 | 0.6031 | 0.6926 | 0.7446 | 0.6964 | 0.6582 | 0.7522 |
| AGUAH | 0.9181 | 0.8089 | 0.7318 | 0.9301 | 1.0000 | 0.8344 | 0.7210 | 1.0080 |
| AMD | 1.0000 | 0.8238 | 0.7300 | 0.9085 | 1.0000 | 0.8301 | 0.7237 | 0.8996 |
| CAASIM | 1.0000 | 0.8967 | 0.8206 | 0.9787 | 1.0000 | 0.9142 | 0.8459 | 0.9886 |
| CAAMTROH | 0.8721 | 0.7779 | 0.7137 | 0.8532 | 1.0000 | 0.9371 | 0.9108 | 0.9742 |
| CAEV | 0.5743 | 0.5191 | 0.4860 | 0.5563 | 0.6757 | 0.6458 | 0.6294 | 0.6698 |
| CMAPS | 1.0000 | 0.8104 | 0.7153 | 0.9049 | 1.0000 | 0.8417 | 0.7671 | 0.8967 |

**Table 10.** *Cont.*

| DMU | CRS Eff Score $\theta_j$ | Bootstrap CRS Eff Score $\hat{\theta}_{jb}$ | Lower Bound $\hat{\theta}_{jb}$ | Upper Bound $\hat{\theta}_{jb}$ | VRS Eff Score $\theta_j$ | Bootstrap VRS Eff Score $\hat{\theta}_{jb}$ | Lower Bound $\hat{\theta}_{jb}$ | Upper Bound $\hat{\theta}_{jb}$ |
|---|---|---|---|---|---|---|---|---|
| CESPT | 1.0000 | 0.8933 | 0.8102 | 1.0809 | 1.0000 | 0.9674 | 0.9377 | 1.0557 |
| COMAPA-R | 0.7840 | 0.7046 | 0.6471 | 0.7677 | 0.8321 | 0.7788 | 0.7376 | 0.8450 |
| COMAPA-EM | 0.7556 | 0.6547 | 0.5940 | 0.7283 | 0.8092 | 0.7512 | 0.7094 | 0.8035 |
| CMAS | 0.4733 | 0.4226 | 0.3879 | 0.4600 | 0.5766 | 0.5447 | 0.5229 | 0.5735 |
| DAPASCH | 1.0000 | 0.8875 | 0.8021 | 0.9971 | 1.0000 | 0.9319 | 0.8740 | 1.0803 |
| JAPAM | 1.0000 | 0.8341 | 0.7206 | 1.0631 | 1.0000 | 0.9148 | 0.8458 | 1.0969 |
| JIAPAZ | 0.6424 | 0.5895 | 0.5573 | 0.6278 | 0.8202 | 0.7839 | 0.7567 | 0.8134 |
| SADM | 0.8720 | 0.7872 | 0.7304 | 0.8779 | 0.9902 | 0.9140 | 0.8530 | 1.0137 |
| SAPASNIR | 1.0000 | 0.8139 | 0.6924 | 1.0091 | 1.0000 | 0.9276 | 0.8670 | 1.0857 |
| SAPACG | 0.8550 | 0.7441 | 0.6684 | 0.8352 | 0.9500 | 0.8540 | 0.7821 | 0.9303 |
| SAPAS | 0.9181 | 0.8290 | 0.7625 | 0.9187 | 1.0000 | 0.8922 | 0.8084 | 1.0818 |
| SACMEX | 0.4821 | 0.4256 | 0.3925 | 0.4623 | 0.6197 | 0.5850 | 0.5643 | 0.6128 |
| SIAPASF | 0.9633 | 0.8716 | 0.8064 | 0.9535 | 0.9809 | 0.9239 | 0.8806 | 1.0224 |
| SMAPA | 0.5033 | 0.4492 | 0.4184 | 0.4832 | 0.5796 | 0.5475 | 0.5326 | 0.5672 |
| SIMAPARG | 1.0000 | 0.8818 | 0.7940 | 0.9791 | 1.0000 | 0.9432 | 0.8941 | 1.0819 |
| SIMAPACO | 1.0000 | 0.8276 | 0.7120 | 1.0434 | 1.0000 | 0.9211 | 0.8558 | 1.0883 |
| SIMAS-A | 0.6814 | 0.6088 | 0.5623 | 0.6601 | 0.6851 | 0.6394 | 0.6098 | 0.6938 |
| SOSAPAMIM | 0.5537 | 0.4987 | 0.4615 | 0.5370 | 0.6737 | 0.6424 | 0.6229 | 0.6684 |
| SOAPAP | 1.0000 | 0.8461 | 0.7390 | 1.0099 | 1.0000 | 0.9208 | 0.8554 | 1.0917 |
| SOSAPAZ | 0.8066 | 0.7222 | 0.6622 | 0.7887 | 0.9637 | 0.9103 | 0.8669 | 0.9702 |
| Min | 0.4733 | 0.4226 | 0.3879 | 0.4600 | 0.5766 | 0.5443 | 0.5201 | 0.5672 |
| Mean | 0.8252 | 0.7249 | 0.6567 | 0.8185 | 0.8802 | 0.8132 | 0.7641 | 0.9021 |
| Max | 1.0000 | 0.8967 | 0.8206 | 1.0809 | 1.0000 | 0.9674 | 0.9377 | 1.0969 |
| Std. dev. | 0.1784 | 0.1455 | 0.1241 | 0.1877 | 0.1512 | 0.1317 | 0.1191 | 0.1760 |

Table 11 shows the results of the bootstrap truncated regression. The CRS bootstrap efficiency was the dependent variable and the five context variables, described in Section 4.3, were the independent variables. The first two variables, water macro-metering and water micro-metering, have a negative coefficient and do not have a significant influence in efficiency scores. The last three variables have a positive estimate and only one showed to have a significant p-value. The number of connections per M3 of water lost has a positive and significant impact on the efficiency scores of the Mexican water utilities. The first positive estimate is wastewater treatment. It is a categorical variable and the result means that water utilities that perform wastewater treatment do not attain higher efficiency scores than utilities that do not perform wastewater treatment. The remaining two positive estimates are continuous variables. The first one is the inverse of volume of water lost per connection and is relabeled as number of connections that account for a cubic meter of water lost. The estimate is positive and significant, which means that the efficiency scores increase as the number of connections increases; it is equivalent to an efficiency increase when the volume of water lost decreases. The last positive estimate is sewer coverage and it does not have any significant impact in efficiency. This result suggests that, in the case of the Mexican water utilities, there is not any significant relationship between sewer coverage and efficiency.

**Table 11.** Results of bootstrap truncated regression.

| Context Factor | Estimate | Std. Error | *t*-ratio |
|---|---|---|---|
| Water macro-measuring | −0.23132 | 0.30854 | −0.74973 |
| Water micro-measuring | −0.10930 | 0.31267 | −0.34956 |
| Wastewater treatment | 0.31406 | 0.26304 | 1.19398 |
| Number of connections per M$^3$ of water lost | 0.69709 | 0.31215 | 2.23321* |
| Sewer coverage | 0.01491 | 0.01023 | 1.45680 |
| Note: *n* = 36 | | | * Significant at 5% |

These results suggest important managerial recommendations for the Mexican water utilities. Firstly, it is highly desirable to have utilities with efficient potable water distribution and efficient wastewater treatment. Figure 3, the main path analysis, identifies wastewater treatment as the arrowhead in the study of water utilities' efficiencies. Hu et al. [35] and Zhou et al. [24] highlight the positive impact of wastewater treatment in the efficiency scores of water utilities. Table 8 shows a mean of 0.5 for the context variable wastewater treatment. This result suggests that about 50% of the Mexican water utilities do not perform wastewater treatment and that, according to the results, it is not a significant factor of efficiency. Therefore, wastewater treatment constitutes an improvement opportunity, it is expected that wastewater treatment could become a factor of efficiency as suggested by the main path analysis (Figure 3). Secondly, leakage in the water system remains as a significant exogenous variable when assessing efficiencies. Mexican water utilities should continue to decrease the volume of water lost per connection. Table 8 shows a high variability of this variable. It is an endless continuous improvement journey that should be part of the daily operational agenda of the water utilities. This result matches the findings of De Witte and Marques [29,38], Ananda [40], Pointon and Matthews [21], and Güngör-Demirci et al. [49]. Thirdly, sewer coverage is not a significant variable that supports efficient performance of the Mexican water utilities. Table 8 reports a mean of 91.91% of sewer coverage, which indicates that most of the water utilities are close to the desirable percentage [21]. However, the minimum is 59%, pointing out an opportunity for some water utilities. Therefore, it is expected that an increase in sewer coverage could have a significant impact in the efficiency of the Mexican water utilities, as reported in other analyzed cases [21].

The results of macro-metering and micro-metering are interesting. Metering is important to maintain control at the different stages of the potable water supply chain. About 80% of the Mexican water utilities carry on at least one type of metering, but only 61% perform both macro-metering and micro-metering. The percentage of macro-metering and micro-metering are the same, but their

allocations are different. Some Mexican water utilities conduct just one of them. It could be expected that better distribution control via metering could help to improve efficiency. However, the results did not support this hypothesis. At the end, differences detected by water metering through the distribution network can only be explained by leakage. As discussed above, leakage has a negative impact on the efficiency of the Mexican water utilities and on others cases as well [21,29,38,40,48]. Minimizing the undesirable impact of leakage could enable an opportunity for better control of water distribution and support the uses of macro-metering and micro-metering across the water supply network.

## 5. Conclusions

This article studied the operational efficiency of Mexican water utilities and the context variables that impact their efficiency. A double-bootstrap DEA method was used. In the first stage, CRS, VRS, and scale efficiencies were computed. Then, a bootstrap approach was used to estimate the true distribution function of the CRS and VRS efficiencies. The corrected scale efficiency was estimated with the corrected CRS and VRS efficiency scores. In the second stage, the corrected CRS efficiency was used as the dependent variable in a truncated regression model. Five context variables were added as independent variables in the regression model: water macro-metering, water micro-metering, wastewater treatment, number of connections per unit of volume of water lost, and sewer coverage. The first three are categorical variables and the last two are continuous variables.

The results showed that only the number of connections per unit of volume of water lost had a positive significant impact on the operational efficiency of the Mexican water utilities. On the contrary, macro-metering, micro-metering, wastewater treatment, and sewer coverage did not have a significant influence on efficiency. These results have managerial implications for these water utilities. Most of the water utilities in Mexico supply potable water and provide sewerage service, but only about 50% of them perform some type of wastewater treatment. Hence, the first important managerial recommendation is to implement wastewater treatment. A significant exogenous variable with undesirable influence in efficiency is leakage. Therefore, water utilities should continue efforts to minimize and control leakage across the distribution network. Lastly, Mexican water utilities should continue the expansion of sewer coverage since it showed a positive estimate on the operational efficiency.

It is hoped that this study will motivate future research on the Mexican water utilities and further investigation on the influence of other context variables in their operational efficiency. Moreover, as highlighted in the main path analysis, future research is needed to understand the relationship between operational water efficiency and wastewater treatment. Other methods for operational efficiency estimation constitute attractive future research streams as well.

**Author Contributions:** All authors listed have contributed substantially to the manuscript to be included as authors. Conceptualization, J.H.A.-R., A.G.C. and E.O.-B.; Methodology, J.H.A.-R.; Software, J.H.A.-R.; Validation, A.G.C., E.O.-B.; Formal analysis, J.H.A.-R., E.O.-B.; Investigation, J.H.A.-R., A.G.C., E.O.-B.; Resources, J.H.A.-R.; Data Curation, A.G.C., E.O.-B.; Writing - Original Draft, J.H.A.-R., A.G.C., E.O.-B., J.Y.S.-G., J.E.N.-R.; Writing - Review & Editing, J.H.A.-R., A.G.C., E.O.-B., J.Y.S.-G., J.E.N.-R.; Visualization, J.H.A.-R.; Supervision, J.H.A.-R., A.G.C., E.O.-B.; Project administration, E.O.-B.; Funding acquisition, J.H.A.-R., A.G.C., E.O.-B. All authors have read and agreed to the published version of the manuscript.

**Funding:** This research received no external funding.

**Acknowledgments:** We appreciate the constructive feedback of two anonymous reviewers who helped us to improve our paper.

**Conflicts of Interest:** The authors declare no conflict of interest.

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
