# Peer review of "Operational Efficiency of Mexican Water Utilities: Results of a Double-Bootstrap Data Envelopment Analysis"

_water, doi:10.3390/w12020553_

Round 1

Reviewer 1 Report

The authors present a topic very relevant to the efficient operations of water utilities. I believe the following comments if addressed will make the paper/article more presentable/useful to readers especially operators/managers of water utilities.

The literature review; specifically the first two sections only describes how the review was carried out (basically a methodology). The authors failed to show/discuss the arguments of the different authors listed and how these are relevant or otherwise to their arguments in this paper. 

Results are well presented, however, authors already start with comparisons/recommendations/conclusions in the analysis section. I think authors should concentrate on clearly explaining the results in the analysis section and present/leave the recommendations for the conclusion section. As this makes it a bit difficult to comprehend what is actually being put across.

A number of inputs/outputs were proposed/used in the efficiency analysis, meanwhile, authors based their conclusions/recommendations on the exogenous context variables specifically wastewater treatment and sewerage. Is there a reason for this? Also, authors provide conflicting statements here, one hand it is mentioned that waste treatment does not lead to higher efficiency scores (line 496, 497) and also wastewater treatment and sewerage coverage did not have a significant influence on efficiency (551), meanwhile the recommendation of a wastewater treatment plant.

Author Response

[1] The literature review; specifically the first two sections only describes how the review was carried out (basically a methodology). The authors failed to show/discuss the arguments of the different authors listed and how these are relevant or otherwise to their arguments in this paper.

We appreciate Reviewer’s comments. We have added a paragraph at each one of the first two sub-sections. They explain the purpose of the lit review sub-section and the role of our paper.

[2] Results are well presented, however, authors already start with comparisons/recommendations/conclusions in the analysis section. I think authors should concentrate on clearly explaining the results in the analysis section and present/leave the recommendations for the conclusion section. As this makes it a bit difficult to comprehend what is actually being put across.

We appreciate Reviewer’s comments. We have changed the name of section 4.4. It is called Results and Discussion now. We want to take the opportunity of analyzing and discussing the results at the same time.

[3] A number of inputs/outputs were proposed/used in the efficiency analysis, meanwhile, authors based their conclusions/recommendations on the exogenous context variables specifically wastewater treatment and sewerage. Is there a reason for this?

We appreciate Reviewer’s comments. Yes, both wastewater treatment and sewerage coverage are in early stages of development in Mexico. Furthermore, water is a finite resource; hence, wastewater treatment plays a critical role in the conservation of this resource. Sewerage is associated with the sanitation services demanded by the population.

[4] Also, authors provide conflicting statements here, one hand it is mentioned that waste treatment does not lead to higher efficiency scores (line 496, 497) and also wastewater treatment and sewerage coverage did not have a significant influence on efficiency (551), meanwhile the recommendation of a wastewater treatment plant.

We appreciate Reviewer’s comments. Most researchers agree that economies of scope are important in the case of water utilities. We think that wastewater treatment and sewerage coverage did not have a significant influence on efficiency because they have not a significant presence in the Mexican water utilities. For example, only about 50% of them perform some type of wastewater treatment. Hence, we expect that if that percentage is increased, then the water utilities could increase their efficiency.

Reviewer 2 Report

Operational Efficiency of Mexican Water Utilities: Results of Double-Bootstrap Data Envelopment Analysis

Water

This paper estimates the efficiency of the Mexican water utilities using the non-parametric method of data envelopment analysis the bootstrap procedure both to give robustness to the results obtained and to investigate the influence of exogenous variables on the scores obtained. The topic of the paper is relevant for the literature and the case study is interesting, however the paper suffers of several issues that need to be improved before the paper is ready for publication.

The first and most important issue is the model specification. The authors do not understand how these models work. First, you cannot use ratios as inputs or outputs, for example you should use volume of water distributed and not the consumption per capita. In this way, the results are biased. Secondly, the inputs or the outputs selection the simple rule to choose them is to think, for example, for an input that the same is its reduction (or for the outputs their expansion). Thus, it is not intended to reduce the accounts with on-time payment and therefore this is not an input.

The second issue and also determinant is the literature review. The authors cannot review the literature considering randomly two or three words and make a set of judgements. The authors should at least include the words efficiency, benchmarking, nonparametric/parametric and water utilities. For example, you include in the review the papers on water efficiency of farms, this is agricultural not urban water. An example, Carvalho, P. has 9 papers published in the web of science concerning efficiency of water utilities and do not is included. A full review of this topic carried out by Berg and Marques published in Water Policy (relative to the year 2009) identified more papers than you (the first paper was published in 60’ and in 70’ and 80’ several papers were published.) Currently the literature, should overcome the 400 papers.

A last major issue are the exogenous variables. Wastewater treatment coverage can not be defined as categorical since the cost is not the same if there is 10%, 50% or 100% of treatment. Also, coverage sewerage can not be an issue (unless of economies od scope purpose) since more coverage, more costs but also more revenues.  

Other minor issues:

Line 64. Add the study of Marques (2008) published in Water Policy about efficiency of private and public water utilities in Portugal;

In Table 7 add the paper published in 2014 in Journal of Water Research Planning and Management about Japanese water utilities and the influence of exogenous variables on efficiency. And the paper of Pinto et al. (2017) published in Urban Water Journal should be included.

Page 237: The true citation is:  DE WITTE, K.; MARQUES, R. (2009). Capturing the environment: a metafrontier approach to the drinking water sector. International Transactions of Operation Research. Vol. 16, no. 2, pp. 257-271

Line 462: 17.18%??? Are you sure an in line 465: 15,42???

Line 493 and next: Replace measuring by metering

Author Response

[1] The first and most important issue is the model specification. The authors do not understand how these models work. First, you cannot use ratios as inputs or outputs, for example you should use volume of water distributed and not the consumption per capita. In this way, the results are biased. Secondly, the inputs or the outputs selection the simple rule to choose them is to think, for example, for an input that the same is its reduction (or for the outputs their expansion). Thus, it is not intended to reduce the accounts with on-time payment and therefore this is not an input.

We appreciate Reviewer’s comments. We understand the issue of using ratios as inputs and outputs. We have added the following paragraph in Section 4.

The sets of inputs, outputs, and context variables are part of the public performance report of the Mexican water utilities. Some of these performance indicators are ratios and their consideration in our research is unavoidable. Firstly, the scarcity of data impedes the transformation of ratios into absolute measures. Secondly, the unbalanced and uneven characteristics of drinking water distribution in Mexico suggest the use of ratio data. For example, if water distribution is intermittent and variable, then the volume of distributed water is a biased measure of consumption of this resource. While on the contrary, the ratio liters/per capita/day provides a better estimation of the volume of water delivered to consumers. Issues associated with ratio data and DEA are discussed by Hollingsworth and Smith [60], Emrouznejad and Amin [61] and Hatami-Marbinia and Toloob [62]. For examples of research articles addressing water utilities efficiency using ratio data, the reader is referred to [12,39,63,64].

We appreciate Reviewer’s comments in regards of accounts with on-time payment. We have extended the discussion of the selection of this indicator as an input.

Zschille & Walter [47] used total revenue as a proxy for total costs. They argued that revenue covers all operational costs. This is true for the Mexican water utilities, which depend on revenue and monetary subsidies from the government for covering operational expenses. We use accounts with on-time payment as proxy for revenue. Furthermore, this approach could be used for reviewing “whether tariffs for water deliveries are reasonable, and if not, by how much they can be reduced.” Zschille & Walter [47, p 3757]. Hence, accounts with on-time payment is the third input and is a proxy of the ratio water-sold/target-population reported by Mbuvi et al. [39]. Since account with on-time payment is the source of income, it is a proxy for revenues [42,43,47,48,49] and capital [21,23,24,38,].

[2] The second issue and also determinant is the literature review. The authors cannot review the literature considering randomly two or three words and make a set of judgements. The authors should at least include the words efficiency, benchmarking, nonparametric/parametric and water utilities. For example, you include in the review the papers on water efficiency of farms, this is agricultural not urban water. An example, Carvalho, P. has 9 papers published in the web of science concerning efficiency of water utilities and do not is included. A full review of this topic carried out by Berg and Marques published in Water Policy (relative to the year 2009) identified more papers than you (the first paper was published in 60’ and in 70’ and 80’ several papers were published.) Currently the literature, should overcome the 400 papers.

We appreciate Reviewer’s comments. Furthermore, we agree with the Reviewer’s comments. We understand that we did not perform a “comprehensive” literature review. We believe a comprehensive literature review is out of the scope of this paper. Therefore, we replaced the word “comprehensive” and we specified that we are performing just a peculiar quick search of related articles. We added a recommendation for the reader interested in a thorough literature survey in the topic.

The changes are as follows:

Therefore, this literature review is split into three sections: the first one is a peculiar analysis of water efficiency studies by means of a bibliometric and main path analysis,…..

…... Notice that this search framework is not exhaustive. It provides just information of papers using either DEA or SFA for assessing water issues. The aim of this section is to present a quick descriptive overview of research activities and trends in the aforementioned search framework. The reader is referred to Berg and Marques [27] for a thorough discussion of a literature survey in the field.

[3] A last major issue are the exogenous variables. Wastewater treatment coverage can not be defined as categorical since the cost is not the same if there is 10%, 50% or 100% of treatment. Also, coverage sewerage can not be an issue (unless of economies od scope purpose) since more coverage, more costs but also more revenues. 

We appreciate Reviewer’s comments. We have added the following clarifications in regards to these context variables.

Several authors identified a significant relationship between efficiency of water utilities and wastewater treatment [30,35,37,38,46]. This synergy suggests the presence of economies of scope [38,42] when a water utility simultaneously distributes drinking water and treats wastewater. Therefore, wastewater treatment is incorporated in this study as a categorical variable; its value is 1 if the water utility performs wastewater treatment, its value is 0 otherwise.

The last context variable is sewer coverage. Zschille and Walter [42] used sewage services as a categorical variable. In this case, sewer coverage is a continuous variable that suggests that companies providing both potable water and sewage services are more efficient than utilities providing just one service, an effect of economies of scope [38]. This idea is aligned with the studies of De Witte and Marques [13,35], Carvalho and Marques [38], Halkos and Tzeremes [1], lo Storto [40] (2013), Pointon and Mathews [20], and Molinos-Senante et al., [30].

Other minor issues:

[4] Line 64. Add the study of Marques (2008) published in Water Policy about efficiency of private and public water utilities in Portugal;

We thank the Reviewer for the recommendation. We added the aforementioned reference.

[5] In Table 7 add the paper published in 2014 in Journal of Water Research Planning and Management about Japanese water utilities and the influence of exogenous variables on efficiency. And the paper of Pinto et al. (2017) published in Urban Water Journal should be included.

We thank the Reviewer for the recommendation. We added the aforementioned references.

[6] Page 237: The true citation is:  DE WITTE, K.; MARQUES, R. (2009). Capturing the environment: a metafrontier approach to the drinking water sector. International Transactions of Operation Research. Vol. 16, no. 2, pp. 257-271

We thank the Reviewer for this observation. We have added the right citation.

[7] Line 462: 17.18%??? Are you sure an in line 465: 15,42???

We thank the Reviewer for this observation. The percentages are wrong. We have added the right values.

[8] Line 493 and next: Replace measuring by metering

We thank the Reviewer for the recommendation. We have replaced measuring by metering.

Round 2

Reviewer 2 Report

The authors incorporate my suggestions/recommendations and the paper improved considerably. Therefore, I recommend its acceptance.